

# Acquiring high-resolution wind measurements by modifying radiosonde sounding procedures

Jens Faber[1,2], Michael Gerding[1], and Torsten Köpnick[1]

[1]Leibniz Institute of Atmospheric Physics at University of Rostock, Kühlungsborn, Germany
[2]now at: Leibniz Institute for Baltic Sea Research, Rostock, Germany

**Correspondence:** Jens Faber (jens.soeder@io-warnemuende.de)

**Abstract.** High-resolved wind measurements are crucial for the understanding of dynamic processes in the atmosphere. In the troposphere and lower stratosphere, radiosondes provide a good spatial resolution of a few meters, but the wind data are usually low-pass filtered by the manufacturer in order to suppress disturbances caused by spurious motions of the sonde. As an example, the filter within the standard processing of Vaisala radiosondes becomes effective at vertical scales below 300 m for

an ascent rate of $5\,\mathrm{ms}^{-1}$.

We describe a method for increasing the usable resolution of radiosonde wind measurements. The main ideas are to avoid self-induced motions of the balloon by keeping it in the sub-critical Reynolds number range, to avoid typical pendulum motions of 15 s period by using a shorter rope, and to use data from a descending balloon in order to avoid disturbances from the wake of the balloon on temperature and humidity measurements due to the decreased rope length. We demonstrate that our changes

in hardware and software allow for artifact free wind data down to scales of 50 m, while remaining disturbances on even smaller scales are removed. Accordingly, the usable resolution of the wind data has been increased by a factor of six compared to the standard data output at relatively low cost.

## 1   Introduction

Radiosondes play an important role for numerical weather prediction in the set of assimilated wind measurements (e.g. Savazzi

et al., 2022). Besides their use in everyday weather forecasting, they are also used for detailed process studies like for example the mountain wave case study by Bramberger et al. (2017). For this type of case studies, there is a measurement gap between standard radiosonde wind observations that are reliable on vertical scales larger than $\sim 300\,\mathrm{m}$ and ultra-fine resolution balloon-borne turbulence measurements (e.g. Söder et al., 2019; Theuerkauf et al., 2011). We propose modifications of standard radiosondes that allow this gap to be bridged while keeping the relative flexibility of radiosonde operation compared

to radars or research aircraft.

While small-scale three dimensional turbulent motions will still need more specialized instrumentation to be measured, we aim to improve measurement capabilities on typical scales of stratified turbulence and below (e.g. Lilly, 1983). This new measurement method will be capable of measuring wind fluctuations at the outer scale of turbulence, i.e. on vertical scales larger than 50 m.



In our study, we concentrate on horizontal wind measurements. Vertical winds can be derived from the ascent rate measurement of the balloon. However, these retrievals are subject to comparatively large measurement uncertainties and exceed the scope of this study (Söder et al., 2019; Wang et al., 2009).

    The general approach for horizontal wind measurements from radiosondes is to use an ascending sounding balloon as a passive tracer. These measurements are affected by several kinds of disturbances. In the past, wind finding accuracy was often

limited by the angular measurement using radio theodolites at the ground to track the radiosonde or its radar reflector (e.g. Houchi et al., 2015). Modern radiosondes, however, are equipped with accurate differential GPS sensors to measure their position. They allow to track the horizontal speed of the radiosonde, resulting in a typical wind speed uncertainty of $0.15\,\mathrm{m\,s^{-1}}$ (Vaisala, 2013). With this low uncertainty, other influences on the wind finding become more important.

    The shape of a sounding balloon in flight is similar to a sphere. This allows to assess the flow conditions around the balloon

from the Reynolds number (Re) based on laboratory experiments as performed by Norman and McKeon (e.g. 2011); Taneda (e.g. 1978); Achenbach (e.g. 1974). Generally, it is assumed that the radiosonde follows the horizontal wind in its specific height. However, for critical and supercritical flow conditions around the balloon it is subject to lateral motions caused by non-symmetrical flow separation around the balloon. These so-called self-induced balloon motions distort the wind measurements and were therefore extensively studied in the 1960s, when the interest in precise stratospheric wind measurements emerged

within the United States' Apollo program (e.g. Scoggins, 1967; MacCready, 1965; Scoggins, 1965).

    Furthermore, movements of the radiosonde below the balloon hamper wind measurement accuracy: The radiosonde is in a coupled system with its carrier balloon. This coupled system acts like a pendulum with a moving pivot. The pendulum motions can be conical or planar, depending on the excitation (Ingleby et al., 2022). They are driven by the above mentioned self-induced motions and by strong vertical wind shears. The frequency of these pendulum motions can be approximated using the

length of the rope between the balloon and the radiosonde: a shorter rope increases the pendulum frequency, thereby allowing for a higher-resolution evaluation of the wind data. On the other hand, temperature and humidity measurements are affected by the balloon's wake (e.g. Luers and Eskridge, 1998; Tiefenau and Gebbeken, 1989), which can be counteracted by lengthening the balloon-radiosonde distance (Söder et al., 2019).

    These different disturbances inhibit the use of radiosonde wind data at spatial resolutions of more than a few hundred

meters. Temporal averaging or spectral smoothing of the data is often applied by modern radiosonde sounding software in order to mitigate the effects for the standard data products. Dirksen et al. (2014) for example use a low-pass filtering with a cut-off period of $40\,\mathrm{s}$ for GRUAN data processing. According to Ingleby et al. (2022), this is similar to the standard RS 92 and RS 41 data processing by Vaisala. By this low pass filtering, wind speed uncertainty is down to $0.15\,\mathrm{m\,s^{-1}}$ as specified for the Vaisala RS41 radiosonde (Vaisala, 2018).

Accordingly, the questions of our study are: How can we mitigate the above mentioned instrumental effects on radiosonde wind measurements without removing the critical scales with a filter? What are the smallest scales where reliable wind measurements are possible?

    To answer these questions, we identify three main factors that limit the accuracy of radiosonde wind measurements on small spatial scales in Section 2. Each of these instrumental effects will be presented alongside counter measures that reduce its





impact on the measurement. This is followed by a presentation of the technical changes to the sounding setup in Section 3 and a discussion of the measures and results in Section 4. In the closing, we present conclusions from our findings.

## 2  How to enhance the accuracy of small-scale horizontal wind measurements on radiosondes

As announced above, we present the three main factors that limit the accuracy of radiosonde wind data on small scales along with counter measures that reduce their impact. They will be demonstrated using two radiosonde launches performed on 28
February 2022 from Kühlungsborn, Germany. The first features all improvements to the sounding setup described in this paper (launched at 11:55 UT). The second is a standard radiosonde Vaisala RS41 SG launched at 15:21 UT. Both launches where conducted using standard radiosonde balloons with a weight of 500 g (Totex TX500). We use Vaisala sondes, because it is the standard system at IAP and they are commonly used by weather services worldwide. However, the effects shown in this study also apply to radiosondes from other manufacturers (e.g. Sippican, GRAW, Meisei).

The measured profile from the standard radiosonde reaches from the ground up to 30.7 km. However, the enhanced radiosonde setup only covers altitudes between 7.7 km and 14.9 km. This was the case, because it has been measured on a descending balloon that is harder to operate and did not reach the intended top altitude of 20 km. Furthermore, we lost radio communications at an altitude of 7.7 km due to obstacles in the line of sight. Nevertheless, we experienced all relevant flow conditions and are able to demonstrate the improvements of the enhanced setup.

In Figure 1 we present zonal and meridional winds from both soundings in the standard resolution as output by the Vaisala Digi-Cora radiosonde receiving station (software: MW41 2.11.0). Temperature and humidity measurements are omitted here, because they will not be discussed further. Both profiles show calm geophysical conditions, with no sharp changes in wind speed except in the planetary boundary layer. The maximum horizontal wind speed does not exceed 13 ms$^{-1}$ in the troposphere and is below 26 ms$^{-1}$ above.

These calm conditions are ideal for our comparison of the two sounding setups: due to the lack of jets, fronts and other mesoscale features we can expect the differences in the small-scales wind spectra between both soundings to be overwhelmingly caused by the technical differences between both sounding setups.

For a technical description of the new radiosonde payload design, please see Section 3.

### 2.1  Create an unfiltered data output

Horizontal wind measurements with radiosondes are done using the sounding balloon as a passive tracer, measuring its horizontal displacement by differential GPS. As presented in the introduction, these measurements are distorted by self-induced balloon motions and by pendulum motions of the gondola carrying the GPS receiver (i.e. the radiosonde). To counter these instrumental effects, wind products offered by major radiosonde manufacturers are usually low-pass filtered (Ingleby et al., 2022; Dirksen et al., 2014).

We demonstrate the effect of this data processing using the conventional ascent of a Vaisala RS41 SG that is described in Figure 2. In this figure, we show zonal wind fluctuations as well as power spectral densities (PSD) of these data. The power





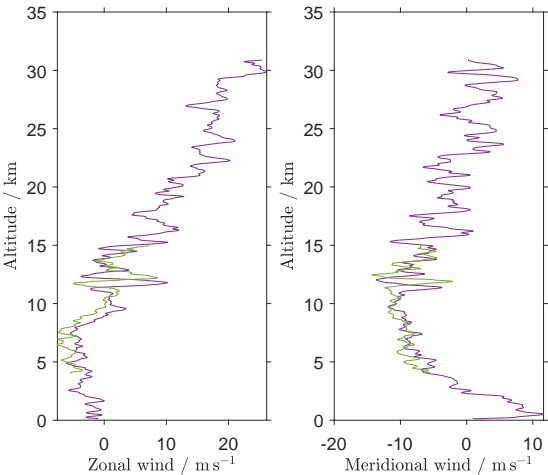

**Figure 1.** Zonal (left) and meridional (right) wind profiles. The purple lines show data from a standard Vaisala RS41 SG radiosonde, the green ones from the enhanced sounding setup presented in this manuscript. Both profiles show the standard output of the Vaisala sounding system.

spectrum has been calculated using the Lomb-Scargle method. This allows the data processing routine to deal with missing data points. The power spectral densities have been Hann-filtered in log-space in order to enhance visibility of certain features.

The altitude plot of the wind data in Figure 2 clearly shows that the standard output lacks higher frequency variations. From the PSD data it is evident that the standard output is increasingly damped at vertical spatial scales smaller than $\sim$300 m (corresponding to time scales smaller than $\sim$67 s).

Therefore, an unsmoothed wind data output is needed for higher-resolution evaluation of the data. We created this by calculating the horizontal drift of the sounding balloon from the raw GPS positions of the radiosonde. A more detailed description for the Vaisala MW41 sounding system is given in Appendix A and shown by a blue line in Figure 2. Please note, that even though we call this the high-resolution data due to the lack of low-pass filtering, both data sets have an output rate of $1\,\mathrm{s}^{-1}$. The unfiltered data in the right panel of Figure 2 clearly show that there is a lot of power in these higher frequencies that are usually damped by the radiosonde receiving system.

For atmospheric fluctuations a -5/3 slope in the power spectrum is expected at these scales smaller than the outer scale of turbulence $L_0$ (up to 200 m, depending on atmospheric conditions, MacCready, 1965). At larger scales, slopes between -2.5 and -3 are expected, with some transition in between (e.g. Nastrom et al., 1997). We show the -5/3 slope in the PSD plots to guide the eye, because we want to highlight positive slopes at smaller frequencies due to instrumental effects. They will be discussed in the following.





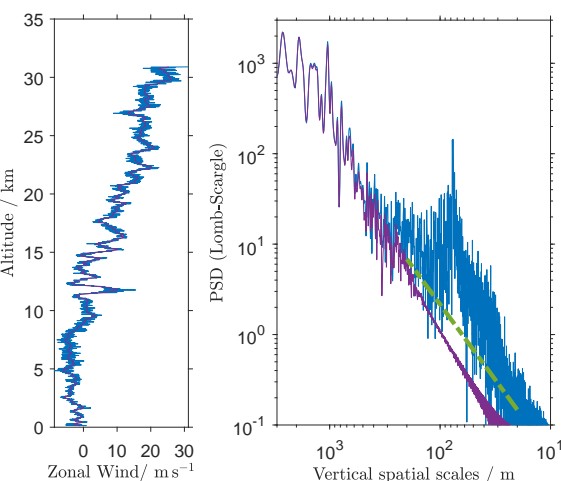

**Figure 2.** Zonal wind measured using the standard radiosonde configuration. Left: raw data, right: Lomb-Scargle power spectral density. Purple lines show standard wind output as given by Vaisala, blue lines show the unfiltered output calculated according to Appendix A. The green line denotes the -5/3 power law.

## 2.2 Avoiding self-induced balloon motions

One of the aforementioned effects are so called self-induced balloon motions. They are random lateral motions of the balloon
that occur only if the flow around the balloon is in the critical or supercritical flow regime (e.g. Murrow and Henry, 1965).

The flow regime around the balloon is best characterized by the Reynolds number:

$$Re = \frac{U\,l}{\nu}, \tag{1}$$

where $U$ describes the characteristic velocity, $l$ the characteristic length scale and $\nu$ the kinematic viscosity of the fluid. In our case, the characteristic length scale is given by the balloon diameter that increases during ascend.

The balloon's s shape during ascent resembles that of a sphere. Therefore, we can rely on extensive studies on smooth and roughened spheres in a uniform flow done by Achenbach (1972) and Achenbach (1974). He describes that the drag coefficient $c_d$ of a sphere depends on the Reynolds number Re of the flow around the sphere. For slightly roughened spheres like an ascending balloon, $c_d$ slightly decreases in the sub-critical Re range (Re < 2.5e5) with increasing Re. For 2.5e5 < Re < 3.5e5, however, $c_d$ decreases sharply with increasing Re (critical Re range).

Consequently, the critical and supercritical Reynolds number range is not only characterized by self induced lateral motions, but also by an unstable ascent rate of the balloon: An increasing ascent rate results in an increase of Re and therefore of $c_d$. The increasing drag slows the balloon down, which reduces Re and $c_d$, accelerating the balloon again. This effect is sometimes called the drag crisis of a balloon.

Figure 3 shows the ascent rates and corresponding Reynolds number for our two test launches. Focusing on the conventional
ascending radiosonde (purple line), we find critical and supercritical flow conditions for altitudes below 12.5 km and sub-





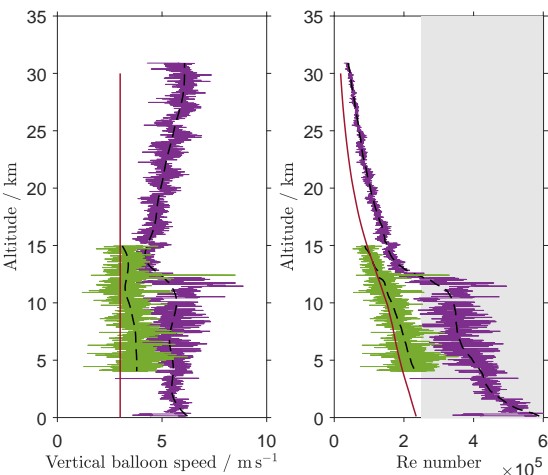

**Figure 3.** Ascent rate (left) and Reynolds number (right) of the flow around the balloon for the standard radiosonde configuration (purple) and the enhanced sounding setup (green) as described in Section 2. The red line shows a simulated ascend of a 500 g balloon with $3\,\mathrm{m\,s}^{-1}$. Dashed lines show low-pass filtered data, the grey shaded area marks the critical and supercritical Reynolds number range according to Achenbach (1974).

critical conditions above. As expected from the flow conditions, we find stronger variations in the ascent rate due to the drag crisis in the lower altitude range. Furthermore, we find an rather sharp reduction in ascent rate from 10 km altitude to 15 km altitude ($5.7\,\mathrm{m\,s}^{-1}$ vs. $4.3\,\mathrm{m\,s}^{-1}$) due to the increased $c_d$ in the sub-critical flow regime. In our case, these changes in vertical balloon velocity are merely used as an indicator that the change of the flow regime is happening as expected from the Reynolds
number.

Here, we will focus on the aforementioned lateral self-induced motions. They occur, because the ring of flow separation on the balloon surface typically deviates from perfect rotational symmetry (e.g. shown for spheres by Taneda, 1978). This induces lateral drag forces on the balloon. Murrow and Henry (1965) visualized this effect by launching a balloon inside a large hangar. Scoggins (1965) showed that the peak in spectral power of this motion is expected for vertical scales of 200 m, with a sharp
decline towards smaller scales and a smoother one towards larger scales.

This effect is perfectly visible on the PSD of the unfiltered data from the normal radiosonde configuration in Figure 4 (blue curve). The figure shows the power spectrum of wind data derived from the standard Vaisala output and the unfiltered data calculated as described in Appendix A. The spectrum is calculated independently for the supercritical Re range (left, below 12.3 km altitude) and the subcritical height range (right, above 12.3 km altitude).
For the unfiltered data, the left panel shows clear deviations from the expected -5/3 slope for atmospheric fluctuations in the inertial subrange on spatial scales smaller than 250 m with a broad peak around 100 m. This peak is caused by self-induced motions of the balloon, which is evident due to its spectral shape well matching the results found by Scoggins (1965, their Figure 9).





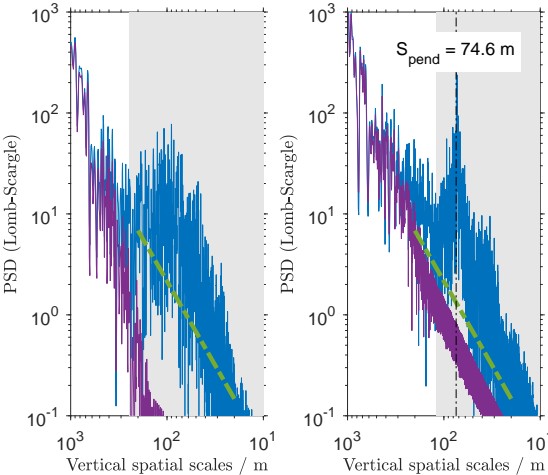

**Figure 4.** PSD for the zonal wind fluctuations of the normal radiosonde configuration from 28 February 2022. Left: Supercritical flow around the balloon (Re > 2.5e5, z < 12.5 km). Right: Subcritical flow around the balloon (Re < 2.5e5, z > 12.5 km). The black dashed line denotes the expected pendulum scale. The blue spectrum is calculated from the unfiltered output, as described in Appendix A. The purple spectrum is calculated from the standard Vaisala output. Grey shaded areas denote spatial scales, where the unfiltered data are impacted by instrumental effects.

The right panel shows the same data from the same flight, except that they are taken in the subcritical Re range. We note in the first place that only data on scales smaller than 120 m show clear deviations from the -5/3 slope. A narrow peak at ∼ 75 m vertical scale can be observed instead of the broad peak in the left panel. Our interpretation of this spectral peak is twofold: First, the self-induced balloon motions are not visible anymore, as expected for the subcritical Re range. Second, a pendulum motion of the radiosonde becomes dominant here, which will be discussed in Section 2.3.

We conclude that keeping the balloon in the subcritical Reynolds number range is a very effective method to avoid self induced motions. In our case, the spatial scales where instrumental effects play a significant role have been roughly halved from 250 m to 120 m.

The simplest way to achieve this is to reduce vertical balloon speeds. Figure 3 shows a simulation of the Reynolds number of the flow around a 500 g balloon at a constant vertical speed of $3\,\mathrm{ms}^{-1}$. It has been calculated using annual averages of temperature and pressure for 50°N from CIRA-86 (Fleming et al., 1990). We find that even at ground level the balloon is expected to be in subcritical flow conditions, eliminating self-induced horizontal motions.





## 2.3 Diminish effects from pendulum motions of the radiosonde

As described in the introduction, a radiosonde below a balloon is subject to pendulum motions. Idealized, these motions can be described by a simple gravity pendulum with a frequency of:

$$f_{\mathrm{P}} = \frac{1}{2\pi}\left(\frac{g}{l}\right)^{\frac{1}{2}}, \tag{2}$$

where $l$ is the length of the rope between the balloon and the radiosonde and $g$ the gravitational constant. For the Vaisala RS 41 with a rope length of 55 m, we expect a pendulum period of 14.9 s. Via the average ascent rate of our standard configuration test launch, this corresponds to a vertical spatial scale of 75 m (black dashed line in Figure 4). This corresponds well to the high-resolution spectrum in the same figure: the narrow peak in the right panel has a center scale of 73 m.

This peak is not clearly visible in the supercritical Re range, even though the flow conditions around the balloon should not

suppress pendulum motions of the radiosonde. We assume that this is due to interactions with the self-induced balloon motions that change the simple gravitational pendulum into one with a moving pivot, shifting its frequency.

These pendulum motions of the radiosonde are forced by vertical shears of the horizontal wind, among other factors. Therefore, they cannot be completely avoided. However, their influence on the wind measurements can be well mitigated by increasing their frequency. This is done by shortening the rope between the balloon and the radiosonde. On our improved payload, we

shortened this rope to 9 m, corresponding to a pendulum period of 6.0 s. As noted in the previous in Section 2.2, we also aim for vertical balloon speeds of 3 ms$^{-1}$ instead of 5 ms$^{-1}$ for aerodynamic reasons. The positive side effect of this measure is to further decrease the center scale of the pendulum peak to vertical scales of 18 m, compared to 75 m for the standard configuration. Furthermore, we attached small wind vanes to the radiosonde with a total area of 0.039 m$^2$ (please see Section 3 for technical details). They dampen pendulum motions, which from our experience otherwise become very vigorous when shortening the

radiosonde rope.

The orange line of Figure 5 shows a PSD of unfiltered data from a flight using the improved payload configuration as it is described below in Section 3. We find a good agreement of the measured power spectrum with the expected -5/3 slope on scales down to 50 m. Assuming a simple gravitational pendulum, the peak is expected at spatial scales of 21 m from the given rope length and ascent rate (black dashed line). There is a well pronounced narrow peak centered at spatial scales of 17.3 m.

As for the subcritical Re range of the standard configuration (right panel of Figure 4), we therefore conclude that this peak is caused by the pendulum motions of the radiosonde. Compared to the standard configuration in the subcritical Re range (right panel of Figure 4), we have been able to further shift the spatial scales of instrumental influences from 120 m down to 50 m.

The light blue line of Figure 5 shows the same data as the orange line but with remaining disturbances from pendulum motions removed by a low-pass filter with a cut-off scale of 40 m. We used a fifth order zero-phase digital Butterworth filter.

Like the standard Vaisala output, this dataset is ready for further geophysical evaluation with all artefacts removed. In contrast, data can be reliably used down to scales of 50 m instead of 300 m.



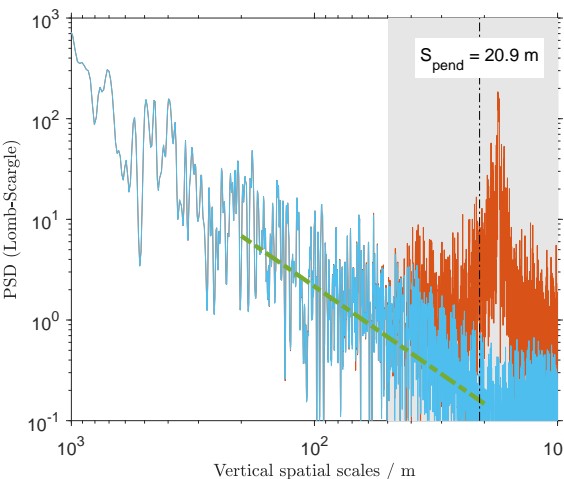

**Figure 5.** PSD calculated from unfiltered zonal wind data on the improved payload as described in Section 3 (orange). The light blue line depicts ready to use wind data from the same payload with the remaining disturbances removed. A -5/3 slope is shown for comparison (green dashed). The black dashed line denotes the expected pendulum scale. Grey shaded areas denote spatial scales, where data are impacted by instrumental artefacts.

## 3 New payload design for high-resolved wind soundings

Compared to a standard radiosonde configuration (e.g. Vaisala, 2018), some distinctive changes have been made for our high-resolved wind soundings. Namely, we decrease the balloon-radiosonde distance to shift the pendulum motions to higher frequencies. This requires measuring on a descending balloon to avoid distortions on temperature and humidity measurements.

The descent speed of the balloon needs to be controlled by an underfilled balloon instead of a parachute. This is, because a parachute cannot procure a constant descend rate. Second, the radiosonde is placed upside down in order to ensure a proper flow around the sensor without flow distortions from the payload. We describe both topics in the following.

We decided to use a two-balloon configuration for the high-resolution wind sounder, similar to the setup used by Kräuchi et al. (2016). Both balloons together (called carrier balloon and descent balloon here) provide sufficient uplift for an ascent rate of $\sim 5\,\mathrm{ms}^{-1}$. The carrier balloon is separated at the top altitude by a pyro cutter that is controlled using custom-made electronics. The pyro cutter in use for our study is model 77003198, type B from TRW Airbag Systems. The uplift of the descent balloon does not suffice for further ascent, but limits the descent speed to $\sim 3\,\mathrm{ms}^{-1}$, depending on the amount of lifting gas. Shortly before touchdown the descent balloon is detached by a second pyro cutter. This prevents the payload from uncontrolled ground drifts after landing. During our test, the cutter additionally opened a piece of cloth in order to slow down the remaining descent and prevent any damage on the ground. Fig. 6 shows this part of the hardware setup.

The electronics has been developed by the authors. The first cutter (terminating the ascent) is released with a timer function. This is preferred to a pressure-sensor, because cost-effective pressure sensors do not have a sufficient resolution at the top





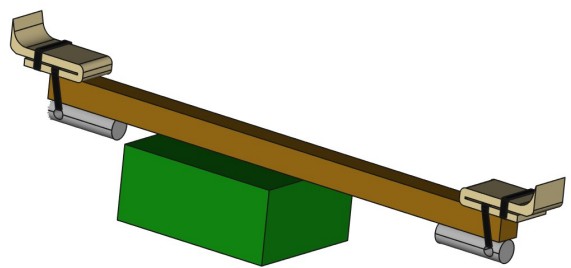

**Figure 6.** Schematic drawing of the balloon mounting and separation devices. This includes the wooden boom (brown), the balloons' necks (sand), pyro cutters (grey) with cable ties (black) and the electronics box (green). Cables and cloth for slowed final descent are removed for clarity.

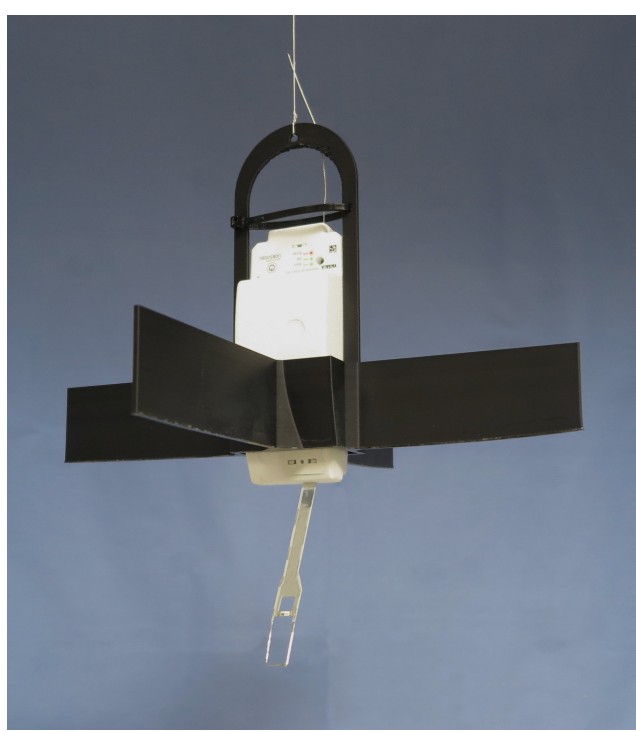

**Figure 7.** Picture of the radiosonde mount for the improved payload design, fixing the sonde upside down 9 m below the balloon.

altitude of the balloon. The second cutter (ending the measurement phase) is triggered closely above ground using an air
pressure sensor. The electronics is shielded by a lightweight 3D-printed box against ambient water and low temperatures.



Both cutters with their respective balloons and the electronics box are mounted on a $\sim 30\,\mathrm{cm}$ wooden stick. This stick avoids uncontrolled touching of both balloons with solid parts of the payload that may damage the balloon hull. The overall weight of this part is 195 g (without balloons).

The radiosonde is mounted in a 3D printed frame 9 m below the balloons as shown in Figure 7. The short cord allows to omit any dereelers. The mount firmly fixes the Vaisala RS41 radiosonde with the sensor boom protruding downward. Little wings reduce the rotation and pendulum motion of the sonde, enhancing the wind data quality at small scales.

For the high-resolution data set, we have calculated the horizontal wind directly from the unfiltered GPS data provided by the Vaisala software. The method for wind calculation is described in Appendix A.

## 4    Discussion

As described in Section 2, we have been able to increase the usable altitude resolution of radiosonde wind measurements from 300 m to 50 m. However, our measures cause higher operational costs and increased complexity. Therefore, we start this section by discussing the cost-benefit ratio of the individual actions.

The retrieval of unfiltered wind data is paramount for any high-resolution wind sounding aiming at vertical scales smaller than 300 m. Furthermore, it does not complicate the operation once it is implemented. Reducing the vertical balloon speed to $3\,\mathrm{ms}^{-1}$ also is a powerful tool without additional cost or complexity: the altitude resolution is more than doubled to less than 120 m due the avoidance of self-induced motions that occur in the supercritical Re range (Figure 4). Presumably, this is further enhanced as the distortions from the pendulum motions are shifted to even smaller scales by the reduced vertical speed. This, however, cannot be shown from this dataset, because it does not feature a standard radiosonde ascending at $3\,\mathrm{ms}^{-1}$.

Further advances in resolution have been made possible by reducing the length of the radiosonde rope from 55 m to 9 m in our case. This further doubles the altitude resolution to 50 m by shifting the pendulum motions to higher frequencies. However, this measure does significantly increase the complexity of the balloon operation: vigorous motions of the radiosonde below the balloon need to be damped by small wind vanes. On an ascending balloon, the radiosonde will almost always be in the wake of the balloon (e.g. Söder et al., 2019). Therefore, a warm bias during daytime and a cold bias during nighttime as well as a wet bias under all conditions would be expected (Kräuchi et al., 2016; Tiefenau and Gebbeken, 1989). Accordingly, combining radiosonde wind measurements with a vertical resolution of 50 m with the humidity and temperature output of the radiosonde is solely meaningful on descent. However, this is only successful, if the descent is slowed by a balloon, as for example suggested by Kräuchi et al. (2016). On a free falling descent, the altitude resolution becomes too low and the high descent rate leads to a warm bias due to frictional heating of the radiosonde (Ingleby et al., 2022; Venkat Ratnam et al., 2014). Using a parachute to slow the descent does not create a sufficiently stable descent rate, because of its dependence on air density and the delayed opening of the parachute (e.g. Ingleby et al., 2022).

We chose to use two balloons and custom made electronics in order to implement the measurement on descent. Another option would be to release lifting gas at the highest point of the flight using a valve (e.g. Hurst et al., 2011; Mastenbrook,



1966). However, these ventilated balloons are much larger than the descent balloon used in our study. This necessitates even lower descent rates to keep them in the subcritical Re range.

In our study, we used comparatively small balloons and low ascent rates to keep the balloon in the subcritical Re range (500 g balloon weight, $\sim 3\,\mathrm{ms}^{-1}$ ascent rate). Another option would be to considerably increase the roughness of the balloon and therefore decrease the critical Re number (Achenbach, 1974). This has been achieved on the so called "Jimspheres": small super pressure balloons with cones on their surface (Scoggins, 1967). However, we do not use this concept, because the "Jimspheres" are not commercially available any more and they cannot achieve a constant vertical speed.

Within this study, we identified and removed several distortions to high resolution evaluations of balloon borne wind soundings. We are able to identify them by their typical deviation in the PSD of the wind fluctuations under calm conditions. Nevertheless, it would be desirable to compare our high-resolution data to an independent measurement. Unfortunately, we are not aware of any technology to achieve this. Radars can achieve high-resolution wind measurements (e.g. Wilson et al., 2014). However, their measurements cannot be compared to balloon borne soundings, due to the spatial inhomogeneity in small scale wind fluctuations and the horizontal drift of the balloon. Other techniques like gliders allow for high horizontal resolutions

(e.g. Wildmann et al., 2021), but they cannot follow the path of a balloon, because they need horizontal airspeed to generate lift.

The description and technical realisation of our study is based on Vaisala RS41 radiosondes that are often used by weather services and research stations. Nevertheless, our method can also be applied to radiosondes of other manufacturers, as all

modern radiosondes measure horizontal winds from the drift of the balloon by GPS.

There are three further effects on balloon wind measurement accuracy that have not been discussed in this study. One of them are internal oscillations of an ascending balloon. They occur on time scales of approximately 2 s (Söder, 2019). They could be avoided using a complicated ballooning scheme suggested by Isom (1949), but they are not relevant at the resolution of 50 m we achieve with the current setup. The second one are inaccuracies of the differential GPS system used on the radiosonde.

Vaisala (2013) finds from a stationary test that the GPS errors on the wind estimate are below $0.05\,\mathrm{ms}^{-1}$, which is below the $0.1\,\mathrm{ms}^{-1}$ resolution used on our data. The third is that the balloon and the radiosonde form a coupled system with the aerodynamic center of pressure located somewhere between the center of the balloon and the radiosonde (e.g. Marlton et al., 2015). Accordingly, the sounding balloon will neither measure the wind at the height of the balloon nor at the height of the radiosonde, but somewhere in between. We cannot assess the uncertainty caused by this affect, but we expect it to be smaller

than on a standard radiosonde, because the balloon-radiosonde distance is reduced from 55 m to 9 m.

In this study, we find the standard radiosonde wind output to be increasingly damped on spatial scales smaller than 300 m. Dirksen et al. (2014) used an effective temporal resolution of the wind data of 40 s in their GRUAN data processing, corresponding to a vertical scale of 200 m for a standard ascent. Ingleby et al. (2022) state that standard radiosonde data processing schemes use similar but not identical cut-off values to Dirksen et al. (2014). Therefore, our findings correspond to the litera-

ture. Unfortunately, the remaining difference cannot be further investigated, as there is no official statement by Vaisala on this matter.



In summary, spatial scales between 50 m and 300 m are accurately resolved by our high-resolved wind measuring method, but neither covered by the standard data products nor by the standard hardware configuration. Nevertheless, our new system is affordable with launch costs in the same order of magnitude as for a standard radiosonde. Furthermore, digital low-pass filtering with a cut-off scale of 40 m removes the remaining artefacts, producing a dataset as easy to use as the standard Vaisala output, but with a usable resolution of 50 m. This can help to bridge the gap between standard radiosondes and highest resolution turbulence measurements from research aircraft or balloons (e.g. Söder et al., 2021; Bramberger et al., 2020; Barat et al., 1984). Therefore, our method allows to improve the outcome of research soundings that are dedicated to high-resolved dynamical phenomena like gravity wave propagation, dynamical instabilities, and tropopause structures including tropopause folds. In a first step, our data have been used for a study that examines kinetic helicity and kinetic energy spectra in flows of stratified turbulence (Dusch et al., 2023, in prep.). Their comparison between analytical theory and atmospheric measurements would not have been possible using a standard radiosonde configuration, because the processes under investigation mainly occur on scales smaller than 300 m.

## 5 Conclusions

Coming back to our question on the achievable altitude resolution from the beginning: using our altered radiosonde operation scheme with the revised payload we can accurately measure wind fluctuations down to vertical spatial scales of 50 m, while not impacting temperature or humidity measurements. This means an increase in usable altitude resolution by a factor of six, because the standard output data are sampled at $\sim 5\,\mathrm{m}$, but filtered below 300 m. Still, our new dataset has all remaining distortions on scales smaller than 50 m removed and is therefore easy to use for further evaluation.

To achieve this result, we retrieved unfiltered position data from the sounding system and mitigated self-induced motions of the balloon and pendulum motions of the gondola. The self-induced motions were avoided by keeping the balloon in the subcritical Re range, i.e. using a comparatively small balloon with a low ascent rate of $\sim 3\,\mathrm{ms}^{-1}$. Pendulum motions where limited using small wind vanes on the radiosonde mount and by reducing the balloon-gondola distance from 55 m to 9 m. This required measuring on a descending balloon in order to retain the data quality of wind and temperature measurements by avoiding the wake of the balloon.

These changes allow for the modified radiosonde to be used in process studies on stratified turbulence. The advantage of our new payload is that it can resolve the outer scale of turbulence and thereby bridge the gap between conventional radiosondes and more specialized turbulence measuring instruments. Nevertheless, it retains the relatively easy and low cost operation and is applicable to all research type radiosonde launches, regardless of the sonde's manufacturer.

*Code and data availability.* The code and data provided with this manuscript can be found at DOI 10.12754/data-2023-0004. During the review process, the content is available at https://owncloud.io-warnemuende.de/index.php/s/rlVg370zkJDCR1g, password: SGK'23review. We provide the following tools for readers who like to implement our suggestions to their sounding setup:



- Data samples: standard Vaisala output and unfiltered position data for both balloon launches described in the manuscript.

- Matlab function to calculate the required balloon lifts depending on desired ascent rate, descent rate and payload.

- Electronics: Board design and firmware

- Mechanics: 3D models of the electronics' housing and the radiosonde mount

The calculations done for the power spectra are well known to the scientific community but the implementation in software very much depends on the user's needs. At IAP, it is embedded in a software solution that is beyond the scope of this study, because it covers all ballooning operations carried out at the institute. Therefore, the code will be made available on request to the authors.

**Appendix A: Wind calculation from unfiltered position data**

The wind calculations is based on the unfiltered (original) GPS data as received by the sounding station. Access to this data stream depends on the system in use. For the Vaisala MW41 Sounding System you can activate a script delivered by Vaisala. The script *RadiosondeLocation.py* continuously records the UTC time, latitude, longitude and geometric height. It can be found in the folder *MW41\ScriptLibrary* of the MW41 installation device (e.g., CD or memory stick). Please, copy the script into the

*SoundingScripts* folder of your PC's harddrive.

To get the GPS position with sufficient precision, we made a small change in the script: In the function *handle_GPSResults* the line

lat = "%10.4f" % (location.PositionWgs84.Latitude)

is changed to

lat = "%12.6f" % (location.PositionWgs84.Latitude).

The same is applied to the variable lon. I.e. the data is recorded with 6 decimals instead of 4. The script is activated in the *Administration / Report Templates and Scripts* section of the MW41 software. The recording during descent needs to be activated in *Administration / Advanced / Sounding*, which should normally be the case for, e.g., RS41. Please refer to the MW41 documentation for details.

The unfiltered position data from the radiosonde are then used to calculate zonal ($u$) and meridional ($v$) wind speeds:

$$u = 2\pi\, r_e\, \frac{cos(\phi)}{360}\, \delta\lambda, \tag{A1}$$

$$v = 2\pi\, r_e\, \frac{1}{360}\, \delta\phi, \tag{A2}$$

where $\lambda$ and $\phi$ are the high resolution longitude and latitude data in degrees, respectively. $r_e$ denotes the Earth's radius and $\delta$ is the central difference operator. From $u$ and $v$ we then calculate wind speed and direction as well as the power spectra shown

above using MATLAB software.

*Author contributions.* JS developed the modifications to the sounding setup, implemented the code for data evaluation as well as plotting, designed mechanical parts of the altered payload and wrote parts of the manuscript. MG contributed to the discussion of the new sounding



setup, oversaw the ballooning operations and wrote parts of the manuscript. TK designed, tested and built the custom made electronics. All contributed to the discussion of the manuscript.

*Competing interests.* The authors declare that they have no competing interests.

*Acknowledgements.* The authors like to thank Michael Priester for his help in preparing and launching the radiosondes as well as an amateur radiosonde hunter, who found our modified payload and sent it back to us for further evaluation. Furthermore, we are thankful to Victor Avsarkisov and Niklas Dusch for their genuine interest in enhanced capabilities as well as technical limitations of the new datasets and for using them in their own studies. Additionally, we are grateful to the MS-GWaves community for countless helpful discussions on high-
resolution and meso-scale measurements.



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
