# Peer review of "Acquiring high-resolution wind measurements by modifying radiosonde sounding procedures"

_EGUsphere, 2023_

## Author Response (AR1)

**Reply on reviewer comment 1 regarding the manuscript egusphere-2023-510**

Our answers to the reviewer comment are listed below with citations from the original comment marked by blue text colour. The changes to the manuscript are highlighted by underlining.

**Summary:**

The manuscript by Söder et al. addresses the vertical resolution of horizontal wind measurements using radiosonde measurements. The operational processing of soundings applies wide filters that reduce the effective vertical resolution to about 300 m. The authors propose several changes to sounding operations, which allow increasing the vertical resolution to about 50 m. They consider the self-induced motion of the balloon in the critical and supercritical flow regimes around the balloon, the pendulum motion of the radiosonde under the balloon, and how to minimize the negative effects their solution could have on measurements of temperature and humidity.

The manuscript is well written, the problem is clearly introduced and described, and the results are well discussed. In particular, the authors discuss the operational costs for each of their solutions, which provide good guidance to those interested in implementing some of their solutions.

I recommend publication of this manuscript after the technical corrections have been implemented, which are mostly a clarification or minor correction of language.

> We are very grateful to the reviewer for the time they spend on our manuscript and for the positive approach of their comments. Furthermore, we are pleased to read that they found our approach to be useful to the science community.

**Technical comments:**

Line 2: Change "spatial" to "vertical".
> Thanks. The change has been made.

Line 23: Delete "measurement".
> The change has been made.

Line 35f: Change semicolons to commas: "… as performed by Norman and McKeon (e.g. 2011), Taneda (e.g. 1978), and Achenbach (e.g. 1974)."
> The handling of references in our manuscript is governed by the Latex template provided by Copernicus. Unfortunately, the use of commas or semicolons as citation separators cannot be chosen by the user.

Line 77: Change "will not be discussed further" to "are not relevant here".
> The suggested change has been made.

Line 86 ff: Shorten "As presented here …" to the end of the paragraph to remove the direct repetition of what was said in the introduction.
> Thanks for the suggestion. The deletion has been done and the beginning of the following paragraph has been amended accordingly.

Line 117: Maybe delete "slightly roughened". I wouldn't call sounding balloons slightly roughened.

The suggested change has been made.

Line 127: " … a rather sharp …"

Thanks. The typo has been corrected.

Line 127: better "… 10 to 15 km altitude …"

We changed to " … from 10 km to 15 km altitude …".

Line 167: Delete "well".

Thanks. The suggested change has been made.

Line 192: Change to "… to ensure proper …"

Thanks. The suggested change has been made.

Line 220: Delete "also".

The deletion has been done.

Line 256: Change "further" to "other".

Thanks. The suggested change has been made.

Line 259: Better write "Inaccuracies of the differential GPS system used on the radiosonde are another effect degrading balloon wind measurements".

Thanks for the suggestions. The change has been made.

**Reply on reviewer comment 2 regarding the manuscript egusphere-2023-510**

We are grateful to the reviewer for the time they have spent on our manuscript. We are especially thankful for pointing out questions about the general validity of our findings concerning different technical and geophysical conditions. In the revised paper, we now address them in Appendix B. Our answers to the individual points raised by the reviewer are given below with citations from the original comment marked by blue text colour. The changes to the manuscript are highlighted by underlining.

General
I found the manuscript interesting and useful.
> Thank you very much.

The data sample, a single pair of profiles, is very small.
Have other similar tests been carried out that would increase the sample?
> Yes, there are another two flights of a very similar setup. They show the same results but have been left out here for clarity. However, they have been used by for a geophysical study on kinetic helicity by Dusch et al., which is currently under revision for Physical Review Fluids.

Are the authors confident that the results are more widely representative (and why)?
> We are very grateful to the reviewer for this and the following questions on the scope of our study. We added Appendix B to the manuscript, where we further discuss the issue.
> In our manuscript we describe general properties of the balloon flight that originate from very fundamental concepts in fluid dynamics and the pendulum motions of the sonde. Accordingly, we are confident that the results are more widely representative, because the effects shown in this study are properties of the flow around the balloon and not of the geophysical conditions. The geophysical conditions have only minor influences through changes in kinetic viscosity. However, our results are not representative of heavier payloads, which in turn lead to larger Reynolds numbers even when using the same balloon type due to the increased amount of lifting gas. As our study focuses on radiosonde soundings, we avoid discussing that in detail in order to improve readability.

One particular aspect is the 'critical/supercritical' conditions up to ~12 km, how typical is this of ascents in mid-latitudes, and other latitudes?
> This is very typical for balloon soundings regardless of latitude and season. This is the case, because the 'critical/supercritical' conditions depend on the Reynolds number, which in turn is a function of air density as the kinematic viscosity is given as $\nu = {}^{\mu}/_{\rho}$. The exponential decay of rho with altitude therefore has the largest influence on Re. The height difference for the critical Reynolds number of the flow around the balloon between the Equator and 50° N is only 900 m (based on CIRA-86). Within the manuscript, this is discussed in Appendix B.

At times I had questions (eg how was the descent limited to 3 m/s in the discussion of Figure 3) which are only addressed later in the manuscript.
> Thank you for pointing this out. We added "For technical information on both soundings, please see Section 3." to the discussion of Figure 3.

Operational radiosonde profiles are necessarily a compromise, and the current string length of 55 m puts more weight on temperature than on wind; I think wind should be given more weight than currently, but a string length of 20 or 25 m (with a small increase in wake effects) is more likely than 9 m. Presumably 20 or 25 m would approximately double the effective resolution of the winds 9with appropriate processing).

> We like to underline that the string length has to be selected depending on purpose. A string length of 20-25 m would shift the center frequency of the pendulum motions only from 75 m to 50 m at 5 m/s. In other words, the wind data are not much improved, but the temperature and humidity data are already compromised. Therefore, for our purpose of high-resolved wind soundings, we prefer an even shorter rope that then requires downleg soundings for avoiding wake effects.

The other 'easy' option would be to reduce average ascent rate from ~5 m/a to say ~4 m/s, how much benefit would this give?

> The answer to this question very much depends on the balloon size that is used. Regarding self-induced motions there is no smooth transition from distorted to non-distorted measurements. Therefore it is vital to keep the balloon in the subcritcal Re range wherever high-resolution wind soundings are required. The influence of the ascent rate and the balloon size on the height of this flow transition is discussed in Appendix B of the revised manuscript.

I have seen very lightweight 'sondes' (eg https://www.skyfora.com/weather-instruments/ even lighter versions may be available in the future) that fall relatively slowly without a parachute - do you envisage that these could be useful for high-resolution wind measurements.

> The skyfora sondes seem very interesting. Compared to a classical balloon based sounding setup, they will stay at subcritical Reynolds numbers even at higher descent speeds due to their small size. Also they avoid pendulum motions by design. For our study however, we focused on widely used sondes. This is also, because free-falling sondes always show a descend rate that strongly decreases with altitude due to the increasing air pressure. A constant altitude resolution can only be achieved by a descent on an underfilled balloon.
> We added the following paragraph to the discussion:
> "Recently, some very lightweight free-falling sondes appeared on the market. They avoid pendulum motions by design and their small size decreases the Reynolds number of the flow around them. However, their descent rate decreases with decreasing altitude due to its pressure dependence. These systems were not available for our study, but it might be worthwhile to compare their results to our approach in another survey."

Detailed comments

1. 'High-resolved' - 'High-resolution' (or 'Highly-resolved') also later
> Thanks very much for pointing this out. We changed it throughout the text.

3. 'suppress disturbances caused by spurious motions of the sonde'
delete 'disturbances caused by'?
> We changed the phrase to 'disturbances on the data caused by [ …]'.

8. 'shorter rope' - cord or string would be better than 'rope'
(rope is usually thicker). I suggest a global change of 'rope' in the manuscript.
> Many thanks for this comment. We changed from 'rope' to 'string' throughout the text. Also, this is in line with the terminology used by Vaisala.

14. 'Radiosondes play an important role for numerical weather prediction in the set of assimilated wind measurements (e.g. Savazzi et al., 2022).'
Savazzi et al. is an interesting paper but doesn't really show the impact of radiosondes in NWP.

A possible alternative is Lawrence et al (2019) they found that 'CONV' data (radiosondes, aircraft* and surface; aircraft are less important in the Arctic than they are in mid-latitudes) have a large impact on Arctic forecasts, especially in winter.

For a study that separates out radiosondes you could look at Candy et al (2021).
Lawrence, H, Bormann, N, Sandu, I, Day, J, Farnan, J, Bauer, P. Use and impact of Arctic observations in the ECMWF Numerical Weather Prediction system. Q J R Meteorol Soc. 2019; 145: 3432-3454. https://doi.org/10.1002/qj.3628
Candy B, Cotton J and Eyre J, 2021:

Recent results of observation data denial experiments

Met Office Weather Science Technical Report 641

https://www.metoffice.gov.uk/research/library-and-archive/publications/science/weather-science-technical-reports
> We are grateful to the reviewer for making us aware of the Met Office report that is a better reference for our introductory sentence. We have changed the reference accordingly.

32. 'They allow to track the horizontal speed' - 'They allow tracking of the horizontal velocity'
> The change has been made.

34. 'This allows to assess' - 'This allows assessment of'
> The change has been made.

36. 'in its specefic height' - 'at each height'
> The change has been made.

37. 'for critical and supercritical flow conditions' please could you briefly explain what this means in non-technical (or minimally-technical) terms.
> Thank you for pointing this out.
> We changed "However, for critical and supercritical flow conditions around the balloon it is subject to lateral motions caused by nonsymmetrical flow separation around the balloon. These so-called self-induced balloon motions distort […]" to "However, for critical and supercritical flow conditions around the balloon it is subject to lateral motions. This is the case, because under these conditions, the flow will not follow the spherical shape of the balloon, but will detach from the balloon downstream of its largest circumference. This detachment is usually not perfectly symmetrical, leading to so-called self-induced horizontal motions of the balloon. They distort […]".

The suggested change has been made.

"[…] spatial resolutions of more than a few hundred meters." has been changed to

"[…] spatial resolutions higher than a few hundred meters." in order to clarify the

unclear wording.

The suggested change has been made.

Thanks for pointing this out. The suggested change has been made.

Thanks, the typo has been corrected.

Thanks for pointing this out. The abbreviation has been replaced by 'our institute'.

We changed to 'from both soundings that were introduced above […]' to make it clear
that we refer to the two soundings that have been introduced above including launch
time and place. We have not calculated the spatial distance between the profiles
because the similarity of the wind profiles is more relevant than their spatio-temporal

distance. Nevertheless, the negligible wind variation within the short time between launches has lead to only small spatial distances.

125. 'we find critical and supercritical flow conditions for altitudes below 12.5 km'

Is 12.5 km (approximately) the tropopause height?

Is it typical to have critical/supercritical conditions in the troposphere and subcritical conditions in the stratosphere?

The tropopause height for this launch was 11.1 km. For radiosonde type balloon launches at 5 m/s ascent rate, the flow condition change usually happens slightly above the tropopause. This is, because the density declines quicker with height above the troposphere, leading to a sharper decline in the Reynolds number (c.f. Figure 3). In the revised manuscript, we discuss the influence of several parameters on the height of the flow change in Appendix B. We added 'For a discussion of the parameters influencing the height of the flow regime change, please see Appendix B.' to the end of the paragraph.

Figure 3. Presumably the 'vertical balloon speed' in green is the descent speed.

Thanks for pointing this out. We changed the wording to 'Absolute vertical speed' in order to include both, the ascent and the descent profile.

Figure 4 caption. 'The black dashed line denotes the expected pendulum scale' add 'given the string length of 55 m' or similar.

Thanks. The suggested change has been made.

170. 'As noted in the previous in Section 2.2' - 'As noted previously in Section 2.2' (or could omit 'previously').

The suggested change has been made.

192. 'a parachute cannot procure' - 'a parachute cannot provide'

Thanks. The wording has been corrected.

292. 'Pendulum motions where' - 'Pendulum motions were'

Thanks. The typo has been corrected.

---

## Author Response (AR2)

**Reply on reviewer report #2 regarding the manuscript egusphere-2023-510 from 21 June 2023**

We are grateful to the reviewer for the time they have spent on our manuscript again. Our answers to the individual points raised by the reviewer are given below with citations from the original report marked by blue text colour.

The resonses were generally helpful. I would like to see some points from them incorporated into the main manuscript.
We are glad that you consider our approach on the revision as helpful. Below we describe how we incorporated more of our answers from the previous revision into the manuscript.

'Two other flights with a similar setup (Dusch et al) show essentially the same results.'
From line 69, we added "Two other soundings of the advanced setup show essentially the same results. Their results are presented in Dusch et al. (2023, under revision) but will be omitted here for clarity."

'In our manuscript we describe general properties of the balloon flight that originate from very fundamental concepts in fluid dynamics and the pendulum motions of the sonde. Accordingly, we are confident that the results are more widely representative, because the effects shown in this study are properties of the flow around the balloon and not of the geophysical conditions.'
This information is now added to the discussion in the paragraph starting at line 228: "In our study, we describe general properties of balloon soundings that originate from fundamental concepts in fluid dynamics and the pendulum motions of the sonde. Accordingly, we are confident that the results are more widely representative, because the effects shown in this study are mainly properties of the flow around the balloon and not of the geophysical conditions. In Appendix B we show the dependence of the height of the flow regime change on sounding setup parameters like balloon size and ascent rate. Furthermore, we briefly discuss the remaining geophysical influences on this."

Detailed comments

8. 'use data from a descending balloon in order to avoid disturbances from the wake of the balloon on temperature and humidity measurements due to the decreased string length.'

Wake effects are more of a problem in the mid-stratosphere - which isn't reached in your descending balloon setup!
My statement is partly based on WMO/Elms et al (1994), they looked at mean temperatures rather than indivudual spikes and recommended "Suspension lengths of less than lOm should be avoided if best quality radiosonde temperature observations are required at pressures lower than 50 hPa. A suspension length·of 40m is probably longer than necessary for most routine operational radiosonde observations." (Of course the balloon diameter is greater at upper levels giving a wider wake. Temperature spikes are usually removed by the ground rocessing.)
We fully agree with the reviewer that wake effects on radiosonde measurements are stronger in the stratosphere due to larger balloon and wake diameters. However, as we

state in the introduction and several other parts of the manuscript, our aim is to increase the altitude resolution of radiosonde data beyond what is achieved on standard soundings. Soeder et al. (2019) estimate that the mean probability for wake encounter on a standard radiosonde ascent is 30 percent (55 m string). Tiefenau and Gebbeken (1989) estimate that the temperature drop in the wake during nighttime at 300 hPa is 0.69 K, which should not be neglected for high-resolution campaigns from our point of view. We added the following sentence at line 235: "These temperature biases in the wake are below one Kelvin around tropopause level and reach above two Kelvin in the stratosphere (Tiefenau and Gebbeken, 1989)."

Specifically for wind research a wind-only GNSS sonde could be used (Graw make one, I think Meteomodem do too). If using a wind-only sonde would you choose to measure on ascent or descent? If high-resolution wind data were required up to say 30 km presumably you would measure on ascent. (Obviously the abstract is not the correct place to discuss all these issues, but I would like to see them discussed somewhere within the manuscript.)

We added the following paragraph to the discussion starting at line 242: "If only high-resolution wind measurements are required for a certain research question, we would suggest using ascending sondes in the sub-critical Reynolds number range with a short string and additional wind vanes. Compared to a descending setup this simplifies the operation without degrading wind data quality."

World Meteorological Organization, 1994: The difference in observed temperatures from radiosondes suspended 10 m and 40 m beneath a 1400 g balloon (J.B. Elms, J. Nash and G. Williams).
Papers Presented at the WMO Technical Conference on Instruments and Methods of Observation (TECO-94), Instruments and Observing Methods Report No. 57, WMO/TD-No. 588, Geneva, pp. 121-126.
https://library.wmo.int/doc_num.php?explnum_id=9607

126. 'sharp reduction in ascent rate from 10 km to 15 km altitude'
Please include the following from your response:
"The tropopause height for this launch was 11.1 km. For radiosonde type balloon launches at 5 m/s ascent rate, the flow condition change usually happens slightly above the tropopause. This is, because the density declines quicker with height above the troposphere, leading to a sharper decline in the Reynolds number (c.f. Figure 3)."

Thanks for your suggestion. The requested addition has been made.

258. 'all modern radiosondes measure horizontal winds from the drift of the balloon by GPS'
'all' - 'most' See section 5.1 of ECMWF Tech memo 807
https://www.ecmwf.int/en/elibrary/80268-assessment-different-radiosonde-types-20152016
Radar is still extensively used by China and Russia (I would describe the Chinese radar sondes as 'modern', but not the Russian ones). Indonesia and some adjacent countries still make extensive use of PILOT ascents, probably radar.

Thanks for pointing this out. The requested change has been made.

289. 'Dusch et al., 2023, under rev.' - 'under revision'

The correction has been made.